Molecular and phenotypic distinctions of macrophages in tolerant and susceptible to hypoxia rats

Dzhalilova Dzhuliia juliajal93@mail.ru 1 2
Kosyreva Anna 1 2
Lokhonina Anastasiya 2
Tsvetkov Ivan 1
Vishnyakova Polina 2 3
Makarova Olga 1
Fatkhudinov Timur 1 2
1 Avtsyn Research Institute of Human Morphology, Petrovsky National Research Centre of Surgery , Moscow , Russian Federation
2 Research Institute of Molecular and Cellular Medicine, Peoples’ Friendship University of Russia named after Patrice Lumumba (RUDN University) , Moscow , Russian Federation
3 National Medical Research Center for Obstetrics, Gynecology and Perinatology Named after Academician V.I. Kulakov of Ministry of Healthcare of Russian Federation , Moscow , Russian Federation
Albertini Maria Cristina
Electronic publication date: 2023 Oct 10
Publication date: 2023
Volume: 11
Electronic Location ID: e16052
Received 2023 May 15; Accepted 2023 Aug 16
Copyright: ©2023 Dzhalilova et al.
Copyright year: 2023
Copyright holder: Dzhalilova et al.
License: This is an open access article distributed under the terms of the Creative Commons Attribution License, which permits unrestricted use, distribution, reproduction and adaptation in any medium and for any purpose provided that it is properly attributed. For attribution, the original author(s), title, publication source (PeerJ) and either DOI or URL of the article must be cited.
License URL: https://creativecommons.org/licenses/by/4.0/

Keywords: Hypoxia tolerance, Inflammation, HIF-1, Rats, Macrophage

Funding: Russian Science Foundation 22-15-00241 Budgetary topic 122030200530-6 “Cellular and molecular biological mechanisms of inflammation in the development of socially significant human diseases” This work was supported by the Russian Science Foundation [grant number 22-15-00241], and budgetary topic 122030200530-6 “Cellular and molecular biological mechanisms of inflammation in the development of socially significant human diseases”. The funders had no role in study design, data collection and analysis, decision to publish, or preparation of the manuscript.

==============================
Individual hypoxia tolerance is a major influence on the course and outcome of infectious and inflammatory diseases. Macrophages, which play central roles in systemic inflammatory response and other immunity reactions, are subject to functional activation orchestrated by several transcription factors including hypoxia inducible factors (HIFs). HIF-1 expression levels and the lipopolysaccharide (LPS)-induced systemic inflammatory response severity have been shown to correlate with hypoxia tolerance. Molecular and functional features of macrophages, depending on the organisms resistance to hypoxia, can determine the severity of the course of infectious and inflammatory diseases, including the systemic inflammatory response. The purpose is the comparative molecular and functional characterization of non-activated and LPS-activated bone marrow-derived macrophages under normoxia in rats with different tolerance to oxygen deprivation. Hypoxia resistance was assessed by gasping time measurement in an 11,500 m altitude-equivalent hypobaric decompression chamber. Based on the outcome, the animals were assigned to three groups termed ‘tolerant to hypoxia’ (n = 12), ‘normal’, and ‘susceptible to hypoxia’ (n = 13). The ‘normal’ group was excluded from subsequent experiments. One month after hypoxia resistance test, the blood was collected from the tail vein to isolate monocytes. Non-activated and LPS-activated macrophage cultures were investigated by PCR, flow cytometry and Western blot methods. Gene expression patterns of non-activated cultured macrophages from tolerant and susceptible to hypoxia animals differed. We observed higher expression of VEGF and CD11b and lower expression of Tnfa, Il1b and Epas1 in non-activated cultures obtained from tolerant to hypoxia animals, whereas HIF-1α mRNA and protein expression levels were similar. LPS-activated macrophage cultures derived from susceptible to hypoxia animals expressed higher levels of Hif1a and CCR7 than the tolerant group; in addition, the activation was associated with increased content of HIF-1α in cell culture medium. The observed differences indicate a specific propensity toward pro-inflammatory macrophage polarization in susceptible to hypoxia rats.

Introduction

Hypoxia tolerance occupies a special place among factors that determine the course of infectious and inflammatory diseases. Apart from serum levels of erythropoietin, corticosterone, norepinephrine and oxidative stress indicators, biochemical correlates of hypoxia tolerance include the antioxidant defense enzymes, heat shock protein and hypoxia-inducible factor (HIF) activities (Ghosh, Kumar & Pal, 2012; Jain et al., 2013; Kirova, Germanova & Lukyanova, 2013; Dzhalilova & Makarova, 2020).

Transcription factors HIF-α and HIF-β mediate reaction of the body to hypoxic conditions. There are three isoforms of HIF-α, designated HIF-1, HIF-2 and HIF-3, and one isoform of HIF-β. Transcription of HIF-dependent genes VEGF (vascular endothelial growth factor), GLUT1 (glucose transporter 1), EPO (erythropoietin), etc., is activated by heterodimers of α- and β-subunits. Under normoxic conditions, α-subunits undergo proteasomal degradation and the activation stops (Semenza, 2012). In animals that are susceptible to oxygen deprivation, HIF-1 expression levels in many organs including the liver and the brain are higher compared with hypoxia-tolerant animals (Jain et al., 2013; Kirova, Germanova & Lukyanova, 2013; Dzhalilova & Makarova, 2020).

Hypoxic responses share many molecular pathways with inflammatory processes, as the latter are accompanied by hypoxia which causes metabolic changes in immune cells (McGettrick & O’Neill, 2020). Individual differences in HIF-1 expression may influence predisposition to local and systemic inflammatory responses. High-altitude pulmonary edema-predisposed individuals have inherently high HIF-1 expression levels (Soree et al., 2016).

Innate immunity reactions are core to any immunological process, including systemic inflammation and such extremes as acute respiratory distress syndrome (ARDS) and cytokine storm (Kosyreva et al., 2021). Macrophages, the key cells of innate immunity, show high phenotypic plasticity: depending on microenvironmental cues, they can polarize into M1 (classically activated, pro-inflammatory) or M2 (alternatively activated, anti-inflammatory). The direction of macrophage activation depends on the nature of inducer: for example, M1 polarization can be driven by microbial component lipopolysaccharide (LPS), whereas M2 polarization can be driven by IL4 (Sica & Mantovani, 2012). M1 macrophages orchestrate pro-inflammatory responses and produce pro-inflammatory factors IL6, IL12 and TNFa, whereas M2 macrophages mitigate the inflammatory reactions and promote healing (Ivashkiv, 2013; Murray et al., 2014). During infectious process, macrophages initially polarize as M1 to fuel the host inflammatory response to pathogens. Sooner or later, the vector of macrophage activation must be reversed in order to preserve the host from its own immunity and restore the normal tissue homeostasis. At certain stages of the process, the M1/M2 balance may have critical significance for the outcome (Schultze, Schmieder & Goerdt, 2015).

HIF-1 supports M1 activation typical of acute phase, to which macrophages adapt through HIF-1-dependent gene expression, but is also involved in M1 macrophage activation under normoxic conditions (Blouin et al., 2004; Frede et al., 2006; Takeda et al., 2010; Tannahill et al., 2013; GalvanPena & ONeill, 2014; Wang et al., 2017). M1 macrophages produce pro-inflammatory cytokines, express surface markers CD80, CD86 and CD16/32, and obtain their energy by HIF-1-controlled glycolysis. HIF-1 deletions interfere with ATP synthesis, thereby influencing the viability, motility and antimicrobial capacity of human and murine macrophages (Cramer et al., 2003; Lin & Simon, 2016; McGettrick & O’Neill, 2020).

HIF-2 positively regulates the arginase-expressing gene Arg1 and supports M2 polarization (Takeda et al., 2010). The alternatively activated macrophages get most of their energy by oxidative phosphorylation, which ensures their long-term involvement in tissue repair. The differential involvement of HIF isoforms in immune response through metabolic reprogramming of macrophages has been confirmed in experimental models. HIF-1, a key mediator of monocyte reprogramming in sepsis, has been considered as a candidate pharmacological target (Shalova et al., 2015).

In our previous studies on rats, the LPS-induced systemic inflammation severity negatively correlated with hypoxia tolerance (Dzhalilova et al., 2019a; Dzhalilova et al., 2019b). Baseline expression of HIF-α isoforms correlates with the individual hypoxia tolerance and may influence macrophage functionalities oxygen-dependently, thereby affecting the course of systemic inflammatory response at the molecular level.

This study aimed at comparative molecular and functional characterization of non-activated and LPS-activated bone marrow-derived macrophages under normoxia in rats with different tolerance to oxygen deprivation.

Materials & Methods

Experimental animals

Male Wistar rats (n = 50), 3–4 months old and weighing 250–300 g, were purchased from the animal breeding facility branch Stolbovaya of the Federal State Budget Institution of Science, the Scientific Center for Biomedical Technologies of the Federal Medical and Biological Agency, Russia. The rats were housed six per 18.5 × 60 × 38 cm cage at regulated room temperature 25 °C ± 2 °C under 12:12 h light–dark cycle and 40–50% relative humidity with unlimited access to water and food (Char, JSC Range-Agro, Turakovo, Russia). All manipulations with animals were carried out according to the European Convention for the Protection of Vertebrate Animals used for Experimental and Other Scientific Purposes (ETS no. 123), Strasbourg, 2006, and all efforts were made to minimize the suffering and distress of animals. Permission was obtained from the Bioethical Committee at Avtsyn Research Institute of Human Morphology (Protocol No. 21 March 29, 2019). The animals served only as a source of the blood; therefore, no experimental conditions and endpoints were applied.

Determination of resistance to hypobaric hypoxia

After 7 days of adaptive feeding, the animals were once placed in a ventilated pressure chamber at an 11,500 m “altitude” and the pressure was lowered at an “ascent” rate of 80 m/s (Jain et al., 2013; Kirova, Germanova & Lukyanova, 2013; Dzhalilova et al., 2019a, Dzhalilova et al., 2019b). Time length till the first sign of characteristic hyperventilatory response (‘gasping time’) was recorded using an electronic stopwatch. To separate animals according to hypoxia resistance, different indicators of gasping time are used, however, the gasping time of tolerant animals should be at least three times higher than of susceptible ones (Mironova et al., 2010; Ghosh, Kumar & Pal, 2012; Jain et al., 2013; Kirova, Germanova & Lukyanova, 2013; Kumar et al., 2014; Dzhalilova et al., 2019a; Dzhalilova et al., 2019b; Kurhaluk et al., 2019; Germanova et al., 2022). Based on gasping time, the animals were assigned to three groups: ‘susceptible’ (<80 s, n = 13), ‘normal’ (80–240 s, n = 25) and ‘tolerant’ (>240 s, n = 12). The ‘normal’ group was excluded from subsequent experiments. Test time was between 08.30 am to 12.30 pm and testing order was randomized daily. For each animal, different investigators were involved as follows: a first investigator (DD) determine the resistance to hypoxia. This investigator was the only person aware of the group allocation. A second investigator (IT) was responsible for the anaesthetic procedure, whereas a third and fourth investigators (AL, PV) performed the cultural procedures, RT-PCR and Western blot.

Isolation of blood monocytes

One month after hypoxia resistance test, the animals were anesthesized (Zoletil, 10 mg/kg) and the blood was collected from the tail vein to isolate monocytes. The samples were mixed 1:1 with HBSS supplemented with heparin (1,000 U/ml, Sintez, Russia). The mononuclear cell fraction was separated by density gradient centrifugation on Ficoll (PanEco, Moscow, Russia) at 400 g and 4 °C for 30 min. The cells were washed twice with HBSS, each time collected by centrifugation at 300 g and 4 °C for 20 min. Cell counts and viability were assessed with a TC20 Cell Counter (Bio-Rad, Hercules, CA, USA).

Macrophage cultures

To obtain non-activated macrophage cultures, the isolated monocytes were transferred to RPMI medium (PanEco, Moscow, Russia) supplemented with 10% fetal calf serum (PAA Lab, Pasching, Austria), 1% penicillin-streptomycin and 50 ng/ml MCSF (Cloud-Clone Corp, Houston, TX, USA) and incubated 24 h at 37°C, 5% CO2 and saturated humidity prior to analysis.

LPS stimulation

For LPS stimulation, the medium with unattached cells was replaced with a fresh portion containing 50 ng/ml LPS E. coli O26:B6 (Sigma-Aldrich, Hercules, CA, USA) on day 2. The cultures were incubated at 37°C, 5% CO2 and saturated humidity for 24 h prior to analysis.

Flow cytometry

Cell surface immunostaining for pan-macrophage markers (CD11b, CD68) and functional state markers (CD163, CD86) used 100 × 103 cells in 100 µl PBS per sample, stained with CD11b-PE (Miltenyi Biotec, Bergisch Gladbach, Germany), CD68-PEVio770 (Miltenyi Biotec, Bergisch Gladbach, Germany), CD86-VioBright FITC (Miltenyi Biotec, Bergisch Gladbach, Germany) and CD163-PE (Thermo Fisher, Waltham, MA, USA) antibodies. The analysis was carried out in a Cytomics FC 500 flow cytometer with CXP software (Beckman Coulter, Brea, California, USA).

RT-PCR

Cell suspensions were immediately fixed with RNAlater RNA Stabilization Reagent (Qiagen, Hilden, Germany), kept at 4 °C for 24 h and transferred to a low temperature freezer for storage at −80 °C. Total RNA was isolated using RNeasy Plus Mini Kit (Qiagen, Hilden, Germany). Estimated RNA concentration in the eluate was 0.1 µg/µl; the quality was controlled by electrophoresis. Randomly-primed reverse transcription of the total RNA was carried out with MMLV RT Kit (Evrogen JSC, Moscow, Russia) at 39 °C for 1 h and the reaction mixture was diluted with 2 volumes of sterile RNase-free water for further use and storage; final dilution of the mixture in PCR constituted 1:250. PCRs were set in duplicates using qPCRmix-HS SYBR (Evrogen JSC, Moscow, Russia) with gene-specific primers in 0.2–0.4 µM final concentrations. The primers were designed using Primer-BLAST online tool and synthesized by Evrogen JSC; the sequences (5′→ 3′): Gapdh f GCGAGATCCCGCTAACATCA, r CCCTTCCACGATGCCAAAGT; Nos2 f CGCTGGTTTGAAACTTCTCAG, r GGCAAGCCATGTCTGTGAC; Hif1a f GAGCCTTAACCTATCTGTCA, r CACAATCGTAACTGGTCAGC; Epas1 f AACCTTAAGTCGGCCACCTG, r TTGCTGTCCAAGGGGATGTC; Vegf f GGATCAAACCTCACCAAAGC, r GGTTAATCGGTCTTTCCGGT; Il1b f CTGTCTGACCCATGTGAGCT, r ACTCCACTTTGGTCTTGACTT; Il6 f TACATATGTTCTCAGGGAGAT, r GGTAGAAACGGAACTCCAG; Il10 f GCCCAGAAATCAAGGAGCAT, r TGAGTG TCACGTAGGCTTCTA; Tnf f CCACCACGCTCTTCTGTCTA, r GCTACGGGCTTGTCACTCG; Tgfb f CCGCAACAACGCAATCTATG, r AGCCCTGTATTCCGTCTCCTT; Mmp9 f ATGGTTTCTGCCCCAGTGAG, r CACCAGCGATAACCATCCGA. Amplification with real time detection and digital analysis of fluorescence was carried out in DTprime Real-time Detection Thermal Cycler (DNA-Technology JSC, Moscow, Russia) with initial heating 95 °C, 5 min followed by 45 × (95 °C, 15 s; 62 °C, 10 s + reading; 72 °C, 20 s). Characteristic values (Cp) for the curves were automatically generated by nonlinear regression analysis and relative expression values were calculated by standard formulas using Gapdh as a reference target (Pfaffl, 2001).

Western blot

Sample preparation and immunoblotting were performed as previously described (Vishnyakova et al., 2021). Briefly, tissue fragments were weighed and lysed in 100 µl of Protein Solubilization Buffer (PSB, Bio-Rad, Hercules, CA, USA) supplemented with complete Protease Inhibitor Cocktail (Roche, Basel, Switzerland), homogenized with microtube pestles and centrifuged at 14,000 g, 4 °C for 30 min. The supernatants were collected, mixed with loading buffer and heated at 95 °C for 1 min prior to 10–12.5% SDS-PAGE separation. The proteins were transferred to PVDF membranes by semi-wet approach using Trans-Blot® Turbo™ RTA Mini LF PVDF Transfer Kit (Bio-Rad). The membranes were blocked with EveryBlot Blocking Buffer for 30 min at room temperature, then stained overnight with primary antibodies to iNOS (ab15323, 1:1000, Abcam, Cambridge, UK), HGF (sc-13087, 1:500, Santa Cruz Biotechnology, Santa Cruz, CA, USA), VEGF (ab46154, 1:1000, Abcam), CCR7 (ab32527, 1:1000, Abcam), GAPDH (sc-25778, 1:1000, Santa Cruz) overnight at 4 °C. The membranes were washed and incubated with horseradish peroxidase (HRP)-conjugated secondary antibodies (Bio-Rad Laboratories, Inc., Hercules, CA, USA) for 1 h at room temperature. The signals were developed using Novex ECL chemiluminescent substrate reagent kit (Invitrogen, Thermo Fisher Scientific, Waltham, MA, USA) in ChemiDoc MP visualization system (Bio-Rad). The protein band optical densities were measured in ImageLab Software with GAPDH as a reference protein.

Statistical analysis

Statistical analysis in Statistica 8.0 included the Kolmogorov–Smirnov normality test followed by Mann–Whitney U test. The differences were regarded statistically significant at p < 0.05. In figures, the data are plotted as medians and IQRs (25–75%), at least five observations in each series.

Results

Immunophenotypes of non-activated and LPS-activated macrophages from tolerant and susceptible to hypoxia rats

Integrin CD11b, is a specific marker of monocyte-derived macrophages widely used in research to verify their identity (Kinoshita et al., 2010; Ikarashi et al., 2013; Nishiyama et al., 2015). As indicated by flow cytometry data, non-activated macrophage cultures from hypoxia-tolerant rats expressed CD11b significantly stronger than corresponding cultures from hypoxia-susceptible rats (Fig. 1). Expression levels of the rest studied macrophage surface markers (CD68, CD86 and CD163) in the two groups of non-activated cultures were similar. Under LPS stimulation, cultures from hypoxia-tolerant animals, but not the susceptible group, reacted by a decrease in CD163 expression. Other macrophage surface markers did not react by changing levels in response to LPS stimulation in both groups.

Figure 1 Flow cytometry for monocytes surface markers applied to non-activated and LPS-activated macrophage cultures obtained from rats with different hypoxia tolerance.

(A) CD11b in non-activated cultures from hypoxia-tolerant rats; (B) CD11b in non-activated cultures from hypoxia-susceptible rats; (C) CD11b, CD68, CD86 and CD163 in non-activated and LPS-activated cultures from tolerant and susceptible to hypoxia rats. Bar heights are medians, whisker ends are upper and lower quartiles; ∗, p < 0.05; ∗ ∗, p < 0.01, Mann–Whitney test.

Expression of pro-inflammatory and anti-inflammatory molecules by non-activated and LPS-activated macrophages from tolerant and susceptible to hypoxia rats

Non-activated macrophage cultures from hypoxia-susceptible rats expressed pro-inflammatory molecules Tnfa (Fig. 2A) and Il1b (Fig. 2B) at significantly higher levels than the tolerant group, whereas baseline expression levels of Il6 (Fig. 2C) in the two groups were similar. Both groups of cell cultures reacted to LPS stimulation by increase in Il1b and Il6 mRNA levels. For Tnfa, a similar increase in mRNA levels under LPS stimulation was observed in cultures from hypoxia-tolerant animals, but not in the susceptible group (Fig. 2A).

Figure 2 Relative mRNA levels of (A) Tnfa, (B) Il1b, (C) Il6, (D) Mmp9, (E) Tgfb and (F) Il10 in non-activated and LPS-activated macrophages from tolerant and susceptible to hypoxia rats.

Bar heights are medians, whisker ends are upper and lower quartiles; *, p < 0.05; **, p < 0.01, Mann–Whitney test.

Non-activated macrophage cultures expressed Mmp9 at similar levels (Fig. 2D). Macrophage cultures from hypoxia-tolerant rats, but not the susceptible group, responded to LPS stimulation by increased expression of Mmp9 mRNA.

Expression levels of anti-inflammatory molecules Tgfb and Il10 were similar between the groups both in resting state and under LPS stimulation. Both groups reacted to LPS stimulation by increase in Tgfb and Il10 mRNA levels (Figs. 2E, 2F).

Nos2 mRNA levels increased under LPS stimulation in both groups, albeit no dynamics in the iNOS protein content were revealed by Western blot analysis (Figs. 3A–3C).

Figure 3 INOS, CCR7, and HGF expression levels in non-activated and LPS-activated macrophages from tolerant and susceptible to hypoxia rats.

(A) Representative Western blots stained with antibodies to iNOS, CCR7, HGF and GAPDH. (B) Relative mRNA Nos2 levels. (C), (D), and (E) Relative levels of iNOS, CCR7 and HGF proteins. Bar heights are medians, whisker ends are upper and lower quartiles; *, p < 0.05; **, p < 0.01, Mann–Whitney test.

As revealed by Western blot analysis, expression levels of M1 marker CCR7 by non-activated macrophages in the two groups were similar. Under LPS, CCR7 protein levels were significantly higher in cultures from hypoxia-susceptible rats, which responded to LPS stimulation by increase in CCR7 protein levels, and no LPS-related dynamics for this marker was observed in the tolerant group (Fig. 3D).

Hepatocyte growth factor, HGF, can direct monocyte migration and differentiation. Monocytic macrophages express both HGF and its receptor c-Met. The low baseline levels of c-Met in non-activated macrophages are known to increase under LPS or IFN-g stimulation (Galimi et al., 2001; Nishikoba et al., 2020). In our setting, macrophage cultures from hypoxia-susceptible rats, but not the tolerant group, responded to LPS stimulation by increased expression of HGF protein (Fig. 3E).

Expression of hypoxia-responsive molecules by non-activated and LPS-activated macrophages from tolerant and susceptible to hypoxia rats

Baseline expression levels of Hif1a were similar between the groups (Fig. 4A); at that, LPS-activated macrophage cultures from hypoxia-susceptible animals expressed Hif1a at significantly higher levels than the tolerant group. Both groups reacted to LPS stimulation by increase in Hif1a mRNA levels (Fig. 4A).

Figure 4 RT-PCR, ELISA and WB results for non-activated and LPS-activated macrophage cultures obtained from rats with different hypoxia tolerance.

(A, C) Relative mRNA levels of Hif1a and Vegf in non-activated and LPS-activated macrophages from tolerant and susceptible to hypoxia rats. (B) HIF-1a protein in culture media conditioned by macrophages from tolerant and susceptible to hypoxia. (D) Relative levels of VEGF protein in non-activated and LPS-activated macrophages from tolerant and susceptible to hypoxia rats. Bar heights are medians, whisker ends are upper and lower quartiles; 7, p < 0.05; 77, p < 0.01, Mann–Whitney test.

HIF-1α protein levels in conditioned cell culture media were similar between the groups in both non-activated and LPS-activated states. Under LPS stimulation, these levels showed significant positive dynamics in cultures from hypoxia-susceptible animals, but not in the tolerant group (Fig. 4B).

Baseline Epas1 (Hif2a) mRNA levels were higher in macrophages from hypoxia-susceptible rats compared with the tolerant group (Table 1) and neither significant responses, nor between-the-group differences in terms of Epas1 mRNA levels were observed under LPS stimulation.

Table 1 Relative mRNA levels of Hif2a in non-activated and LPS-activated macrophages from tolerant and susceptible to hypoxia rats.

Me (IQR); p, Mann–Whitney test. Significant values (p < 0.05) are in bold.

	Epas1 (Hif2a)	p-level	
	Tolerant	Susceptible		
Non-activated macrophages	0.0002
(0.00008–0.0004)1	0.0004
(0.0004–0.0005)2	p 1−2 = 0.03	
LPS-activated macrophages	0.0003
(0.0001–0.1567)3	0.008
(0.0007–0.0250)4	p3−4 = 0.20	
p-level	p1−3 = 0.99	p2−4 = 0.20		

Baseline Vegf/VEGF mRNA and protein levels were significantly higher in macrophages from hypoxia-tolerant rats compared with the susceptible group. In cultures obtained from hypoxia-susceptible rats, Vegf mRNA levels increased under LPS stimulation, but no dynamics were observed for VEGF protein. In the tolerant to hypoxia animals, Vegf/VEGF levels mRNA and protein levels showed no LPS-related dynamics (Figs. 4C, 4D).

A summary of the data on expression of macrophage markers and hypoxia/inflammation-related molecules by non-activated and LPS-activated macrophages from tolerant and susceptible to hypoxia rats is given in Table 2.

Table 2 Research summary.

Parameter	Tolerant,
non-activated vs LPS-activated macrophages	Susceptible,
non-activated vs LPS-activated macrophages	Dif. tolerant vs susceptible	
Immunophenotype	
CD11b+	no dynamics	no dynamics	lower in non-activated macrophages from hypoxia-susceptible rats	
CD68+	no dynamics	no dynamics	no	
CD86+	no dynamics	no dynamics	no	
CD163+	↓	no dynamics	no	
Pro-inflammatory molecules	
Tnfa	↑	no dynamics	higher in non-activated macrophages	
Il1b	↑	↑	from hypoxia-susceptible rats	
Il6	↑	↑	no	
Nos2	↑	↑	no	
Mmp9	↑	no dynamics	no	
CCR7	no dynamics	↑	higher in LPS-activated macrophages from hypoxia-susceptible rats	
HGF	no dynamics	↑	no	
Anti-inflammatory molecules	
Il10	↑	↑	no	
Tgfb	↑	↑	no	
Hypoxia-responsive molecules	
Hif1a	↑	↑	higher in LPS-activated macrophages from hypoxia-susceptible rats	
Epas1
(Hif2a)	no dynamics	no dynamics	higher in non-activated macrophages from hypoxia-susceptible rats	
HIF-1	no dynamics	↑	higher in LPS-activated macrophages from hypoxia-susceptible rats	
Vegf	no dynamics	↑	lower in non-activated macrophages from hypoxia-susceptible rats	
VEGF	no dynamics	no dynamics	lower in non-activated macrophages from hypoxia-susceptible rats	

Discussion

Activated macrophages are classified as pro-inflammatory M1 and anti-inflammatory M2. Induced by Toll-like receptor ligands (bacterial LPS) or Th1 cytokines (TNFα, IFNγ, CSF2), the M1 phenotypes are marked by surface expression of TLR2, TLR4, CD80 and CD86 (Van Dalen et al., 2018; Wang et al., 2019b). M1 macrophages exert high antigen-presenting capacity (Biswas & Mantovani, 2010). They produce reactive oxygen species and pro-inflammatory cytokines IL-1, IL-6, IL-12, IL-18, IL-23 and TNFα that modulate Th1-mediated antigen-specific inflammatory reactions (West et al., 2011; Murray et al., 2014; Zheng et al., 2017). High counts of M1 macrophages in cancers have been associated with favorable prognosis (Mills, 2012; Honkanen et al., 2019). iNOS, a key marker of M1 macrophages, facilitates production of NO from L-arginine (Yao, Xu & Jin, 2019).

The M1/M2 concept is a convenient simplification, since macrophages constitute dynamic cell populations continuously tuning gene expression signatures under microenvironmental cues. Experimental studies of macrophage phenotypes and functional properties in vitro and in vivo conventionally ignore the high individual variability in hypoxia tolerance at the body level. In this study, we addressed this factor in rat model and observed its significance. Non-activated macrophage cultures isolated from Wistar rats with high hypoxia tolerance expressed higher levels of integrin CD11b and VEGF proteins and Vegf mRNA and lower levels of Il1b, Tnfa and Epas1 mRNA than identical cultures from rats with low hypoxia tolerance. All cultures reacted to LPS by increased expression of Il1b, Il6, Nos2, Tgfb, Il10 and Hif1a mRNA; at the same time, the activation promoted a decrease in CD163 protein and an increase in Tnfa and Mmp9 mRNA expression specifically in cultures from hypoxia-tolerant rats, whereas the susceptible animals reacted by increased synthesis of HGF protein and Vegf mRNA. The differences between activated cultures included higher levels of CCR7 and HIF-1 proteins and Hif1a mRNA in macrophages obtained from animals with low hypoxia tolerance.

Integrin CD11b is an accepted marker of bone marrow-derived macrophages, as monocytic lineages isolated from the bone marrow usually express it at high levels in primary cultures (Kinoshita et al., 2010; Ikarashi et al., 2013; Nishiyama et al., 2015; Hoeffel & Ginhoux, 2018). The observed immunophenotypic differences between bone marrow-derived macrophages of rats with high and low hypoxia tolerance included stronger positivity for CD11b in resting state and a negative dynamics for CD163 under LPS stimulation in the former. In cultures obtained from susceptible to hypoxia animals, none of the studied molecules revealed LPS-related dynamics.

Schif-Zuck et al. (2011) identified a new subset of macrophages in mice, emerging upon resolution of peritonitis and expressing relatively low levels of CD11b (Schif-Zuck et al., 2011). These pro-resolving CD11blow macrophages had unique expression profiles, comprising lower levels of M1 enzymes COX2 and MMP9 than their CD11bhigh counterparts, and neither iNOS nor Arg1. These cells are converted from CD11bhigh macrophages which engulf apoptotic neutrophils at the site of inflammation while expressing both M1 and M2 markers including iNOS, Arg1, COX2 and MMP9. Once they engulf ‘enough’ apoptotic cells (estimated 7), they convert into CD11blow cells, switch on 12/15-lipoxygenase to produce the pro-resolving lipid mediators and emigrate to lymphoid organs to convey resolution signals to lymphocytes (Schif-Zuck et al., 2011). The authors emphasize the role of saturated efferocytosis as a signal that generates CD11blow macrophages essential for the containment of inflammatory agents and acute phase termination. The CD11blow macrophages form distinct F4/80+ macrophage subpopulations in lymph nodes and the spleen, exceeding the local CD11bhigh cells in number by contrast with the peritoneum where they originate, which indicates the necessity of the CD11bhigh-to-low conversion for the transfer. These findings illustrate the tunable nature of CD11b positivity and its functional relevance.

LPS is a potent inducer of M1 macrophage phenotypes producing pro-inflammatory cytokines TNFα, IL-1 and IL-6 (Mills et al., 2000; Mantovani et al., 2013). Cheap and effective, LPS is often used to activate macrophages in experimental models (Zhang et al., 2015; Park et al., 2015). In our setting, expression of pro-inflammatory cytokine genes Il1b and Tnfa in non-activated macrophages was significantly higher in cultures obtained from susceptible to hypoxia rats. Despite its prominent links with M2 activation, transcription factor HIF-2α is known to partially control IL-1β production typical of M1 phenotypes (Tannahill et al., 2013). Significantly higher Epas1 mRNA levels in non-activated macrophages obtained from susceptible to hypoxia animals coincided with higher Il1b mRNA levels. Under LPS stimulation, Il1b mRNA levels increased in all studied macrophage cultures, whereas Tnfa mRNA levels increased specifically in cultures obtained from hypoxia-tolerant animals. Il6, Il10 and Tgfb were expressed at similar levels in all non-activated macrophage cultures; under LPS stimulation, similar increases in expression of these genes were observed in both groups.

The inducible nitric oxide synthase (iNOS, or Nos2), a key nitric oxide (NO)-producing enzyme of inflammatory response. At the same time, iNOS has a deterring influence on M1 activation, mediated by NO and combined to its being a key marker of M1 macrophages with a prominent role in pathogen killing. Moreover, this deterring influence has been shown to come from within, through dedifferentiation of M1 macrophages self-promoted by the expression of iNOS (Xue et al., 2018). These findings illustrate the potential of innate immune system in containment of its own homeostasis and the immunological equilibrium locally and systemically. Apart from macrophages, the Nos2 gene is expressed in multiple cell types including dendritic cells, NK cells and primary tumor cells (Bogdan, 2001). Nos2-deficient mice are highly susceptible to inflammatory diseases (Niedbala et al., 2011; Mao et al., 2013). In models of LPS-induced immunological shock, iNOS deficiency aggravates the condition through excessive M1 macrophage activation. In tumors, iNOS inhibition promotes M1 macrophage differentiation resulting in tumor regression (Xue et al., 2018). We observed no hypoxia tolerance-related differences in iNOS expression by macrophage cultures. Though all cultures reacted to LPS stimulation by increase in Nos2 mRNA levels, no corresponding dynamics were revealed for iNOS protein by immunoblotting. Transcription and translation of any protein is a dynamic process. According to Lu et al., (2009) the iNOS mRNA and protein levels increased considerably at 8 h after LPS treatment of Raw 264.7 cells, and based on Wang et al. (2019a) data –after 12 h. Prolonged iNOS upregulation will generate high-level NO which may cause tissues damage, thus efficient degradation of iNOS is a critic mechanism for elimination of inflammation (Golde et al., 2003). According to Lu et al. (2015) NO significantly suppresses iNOS expression at the transcriptional level and had no noticeable effects on iNOS protein stability. iNOS has been revealed to be degraded by ubiquitin-proteasome system (Wang et al., 2018) and autophagy, in a selective manner via the interaction with p62 (Wang et al., 2019a). We evaluated the iNOS protein level and mRNA expression 24 h after LPS stimulation of macrophages, and most likely recorded the time point of the next activation of Nos2 mRNA expression, which was caused by a decrease in NO production. Moreover, it is not expected that induced transcription immediately leads to increased protein levels because maturation, export, and translation of mRNA take some time. Thus, there is a delay between transcriptional induction and protein level increases (Liu, Beyer & Aebersold, 2016).

The LPS-induced macrophage activation results in metabolic switch to glycolysis and pentose phosphate pathway, which stabilizes HIF-1α protein by the same mechanism as in hypoxia (Tannahill et al., 2013; Wang et al., 2017). Driven by multiple arms of NF-κB signaling, HIF-1α expression accompanies and supports the massive production of pro-inflammatory cytokines, glycolytic enzymes, glucose transporters and other M1 effector molecules. By contrast, HIF-2α expression is NF-κB-independent, consistently with its role in the alternative macrophage activation (GalvanPena & ONeill, 2014). In our setting, macrophage cultures obtained from susceptible to hypoxia animals expressed higher levels of Hif2a when non-activated and higher levels of HIF-1 when LPS-activated. A more substantial increase observed for HIF-1 in LPS-activated macrophage cultures from hypoxia-susceptible rats indicates their higher pro-inflammatory aptitude. These data are consistent with increased severity of systemic inflammatory response including more pronounced inflammatory changes to lung and liver tissues in hypoxia-susceptible Wistar rats (Dzhalilova et al., 2019a; Dzhalilova et al., 2019b).

Despite a considerable functional overlap between HIF-α isoforms, they are not capable of reciprocal compensation in knockdown experiments indicative of their different biological purposes (Fang et al., 2009). HIF-1 protein levels in brains, livers and other organs of rats negatively correlate with hypoxia tolerance. Developmentally, HIF-2 upregulates several master genes involved in angiogenic switch, destabilization and vessel sprouting events through recognition of hypoxia-response elements (HREs) in their promoters or enhancers (Befani & Liakos, 2018). The main angiogenic factor VEGF and its receptors VEGFR1 (Flt1) and VEGFR2 (Flk1) are directly induced by HIF-2 under hypoxic conditions (Elvert et al., 2003; Takeda et al., 2004). HIF-2 has been attributed with a stronger transactivation capacity toward VEGF promoter than HIF-1 (Ema et al., 1997; Rankin et al., 2008). In our setting, VEGF expression at both mRNA and protein levels was higher in non-activated macrophage cultures from hypoxia-tolerant rats compared with the susceptible group. However, Hif2a expression was higher in non-activated macrophage cultures from susceptible-to-hypoxia rats. Accumulation of HIF-2α is predominantly due to post-translational regulation as mRNA levels are not significantly induced under hypoxia (Wiesener et al., 2003). Probably, in susceptible to hypoxia rats, the expression of Hif2a mRNA under normoxic conditions is higher than in tolerant, however, the HIF-2α protein is subjected to proteasomal degradation by prolyl hydroxylases (Patel & Simon, 2008) and does not lead to an increase in the VEGF level. In tolerant to hypoxia rats, an increased VEGF expression level under normoxic conditions may be due to other mechanisms. For instance, it was demonstrated (Riazy, Chen & Steinbrecher, 2009), that oxidized LDL can activate VEGF expression in macrophages. We have previously demonstrated that the phagocytic activity of monocytes is higher in tolerant to oxygen deficiency animals under hypoxic conditions (Dzhalilova et al., 2023). It is possible that greater phagocytic activity of monocytes in tolerant animals contributes to greater uptake of oxidized LDL and VEGF activation. Under LPS, macrophage cultures from hypoxia-susceptible rats reacted by increase in VEGF expression at both mRNA and protein levels, paralleled by increase in HIF-1 expression at both mRNA and protein levels. By contrast, cultures obtained from hypoxia-tolerant rats showed no significant dynamics in VEGF or HIF-1 expression under LPS stimulation.

CCR7 or CD197, another marker of M1 polarization, is involved in memory T cell homing and can stimulate dendritic cell maturation (Bono et al., 2007; Tateyama et al., 2009; Woodland & Kohlmeier, 2009). The onset of inflammation is marked by expression of two CCR7-specific ligands, CCL19 and CCL21 (Zlotnik & Yoshie, 2000; Badylak et al., 2008; Oh et al., 2012). The CCR7 gene, which contains HRE and is regulated by HIF complexes, has been suggested as a mediator of pro-metastatic effects of HIF-1α and HIF-2α in non-small cell lung cancer (Wilson, Burchell & Grimshaw, 2006; Li et al., 2009). In our setting, LPS promoted an increase in CCR7 protein specifically in macrophages obtained from hypoxia-susceptible rats, accompanied by an increase in HIF-1α expression at both mRNA and protein levels. As shown by us previously, the LPS-induced immune responses in hypoxia-susceptible Wistar rats proceed with increased involvement of humoral mechanisms (Dzhalilova et al., 2019b). Considering the emerging critical role of CCR7 in the dendritic cell-independent B cell activation, the observed immunological distinction may involve a CCR7-dependent component (Qi et al., 2006; Scandella et al., 2007).

Hepatocyte growth factor (HGF), a multifunctional secreted protein of mesenchymal origin, confers mitogenic, morphogenic and angiogenic effects (Bolanos-Garcia, 2005; Nakamura & Mizuno, 2010). The binding of HGF to its receptor c-Met can trigger Hif1a expression in PI3K-dependent mode in HepG2 hepatoma cells (Tacchini et al., 2001). M2, but not M1, tumor-associated macrophages not only promote proliferation, colony formation and migration of hepatoma cells but also significantly confer tumor resistance to sorafenib via sustaining tumor growth and metastasis by secreting HGF. HGF activates HGF/c-Met, ERK1/2/MAPK and PI3K/AKT pathways in tumor cells (Dong et al., 2019). In our setting, macrophage cultures from hypoxia-susceptible rats, but not the tolerant group, responded to LPS stimulation by increased expression of HGF protein, which under in vivo conditions could promote the triggering of ERK1/2/MAPK and PI3K/AKT signaling pathways in cells and tumor growth.

Matrix metalloproteinase 9 (MMP9) participates in inflammation, angiogenesis and regenerative remodeling; it has been featured as a disease progression/severity marker (Ardi et al., 2009; Deryugina & Quigley, 2010; Kessenbrock, Plaks & Werb, 2010; Al-Batran et al., 2012; Chu et al., 2012; Gelzo et al., 2022). The prominent role of MMP9 in extracellular matrix turnover is combined to its modulatory involvement in the synthesis, release and proteolytic activation of cytokines and chemokines (Davey, McAuley & O’Kane, 2011; Nissinen & Kahari, 2014). MMP9 expressed by neutrophils and macrophages facilitates immune infiltration through destruction of basement membrane components (Zeng, Cohen & Guillem, 1999; Reif et al., 2005; Greenlee et al., 2006; Lin et al., 2008; Luchian et al., 2022). Induction of MMP9 by pro-inflammatory stimuli including LPS interferes with cell signaling and inhibits tissue repair (Leppert et al., 2000; Hu et al., 2007; Yang et al., 2012). Although HIF-1α can indirectly activate MMP9 (Wan et al., 2011), in our setting, Mmp9 expression in macrophages from hypoxia-susceptible rats did not react to LPS stimulation despite the more pronounced LPS-induced increase in HIF-1α expression compared with the tolerant group.

Correction of homeostasis through modulation of macrophage-mediated responses in affected tissues is an important field in biomedicine; the possibilities involve in situ or ex vivo macrophage reprograming (Poltavets et al., 2020; Korzhenevsky & Korzhenevskaya, 2022). Despite the well-developed concepts of macrophage heterogeneity, the relationship of macrophage phenotypes with local oxygenation levels combined to individual hypoxia tolerance remains unexplored. This study for the first time specifies the molecular and phenotypic distinctions of macrophages in rats sharply differing by individual hypoxia tolerance. The data indicate a specific propensity toward pro-inflammatory macrophage polarization in hypoxia-susceptible animals.

Conclusions

Thus, for the first time, the various molecular and phenotypic characteristics of macrophages in animals with different resistance levels to hypoxia are demonstrated here. We observed higher expression of VEGF and CD11b and lower expression of Tnfa, Il1b and Epas1 in non-activated macrophage cultures obtained from tolerant to hypoxia animals, whereas HIF-1α mRNA and protein expression levels were similar. LPS-activated macrophage cultures derived from susceptible to hypoxia animals expressed higher levels of Hif1a and CCR7 than the tolerant group; in addition, the activation was associated with increased content of HIF-1α in cell culture medium. The observed differences indicate a specific propensity toward pro-inflammatory macrophage polarization in susceptible to hypoxia rats. Specific features of macrophages in tolerant and susceptible to hypoxia rats can become the basis of new personalized approaches for inflammatory diseases treatment in accordance to individual hypoxia resistance.

Supplemental Information

Supplemental Information 1 Raw data

Click here for additional data file.

Supplemental Information 2 Uncropped membranes

Click here for additional data file.

Supplemental Information 3 ARRIVE 2.0 Checklist

Click here for additional data file.

Additional Information and Declarations

Competing Interests

Author Contributions

Animal Ethics

Data Availability

The authors declare there are no competing interests.

Dzhuliia Dzhalilova conceived and designed the experiments, performed the experiments, analyzed the data, prepared figures and/or tables, authored or reviewed drafts of the article, and approved the final draft.

Anna Kosyreva conceived and designed the experiments, analyzed the data, authored or reviewed drafts of the article, and approved the final draft.

Anastasiya Lokhonina conceived and designed the experiments, performed the experiments, prepared figures and/or tables, and approved the final draft.

Ivan Tsvetkov performed the experiments, prepared figures and/or tables, and approved the final draft.

Polina Vishnyakova performed the experiments, analyzed the data, authored or reviewed drafts of the article, and approved the final draft.

Olga Makarova analyzed the data, authored or reviewed drafts of the article, and approved the final draft.

Timur Fatkhudinov conceived and designed the experiments, authored or reviewed drafts of the article, and approved the final draft.

The following information was supplied relating to ethical approvals (i.e., approving body and any reference numbers):

Bioethical Committee at Avtsyn Research Institute of Human Morphology provided full approval for this research (Protocol No. 21 March 29, 2019).

The following information was supplied regarding data availability:

The raw data are available in the Supplemental Files.

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
