# Peer review of "Molecular and phenotypic distinctions of macrophages in tolerant and susceptible to hypoxia rats"

_PeerJ, doi:10.7717/peerj.16052_

## Round 0.1 · original submission · Major Revisions

Please, follow Reviewers suggestions.

·

Basic reporting

In the manuscript titled " Molecular and phenotypic distinctions of macrophages in tolerant and susceptible to hypoxia rats (#85424)", Dzhalilova et al. provide a molecular and phenotypic characterisation of macrophages in rats with different resistance levels to hypoxia. This is an interesting study on key factors contributing to macrophage response and polarisation. The results would be of great interest to the field of immune response.
There are a few points that the authors need to clarify before their paper is suitable for publication.

1. For the clarity of reading could the authors please consider explaining “the other group” statement in the manuscript. There are different comparisons made for the four groups and therefore it is often less clear for the reader, which of the groups is meant when it is referred to as the other study group. Example being: line 219: “Non-activated macrophage cultures from hypoxia-susceptible rats expressed Tnfa and Il1b at significantly higher levels than the other group”. This could state: “ than the tolerant group.”
2. Could the CCR7 data be presented in one figure, ie pull together Fig 3 and Fig 5? Otherwise, CCR7 relevant data is presented in different figures.
3. Similarly, could table 3 be added to figure 7, or both presented as figures? Otherwise, HIF1a relevant data is presented in different places.
4. Could authors please comment on the whiskers ends being generally higher for the LPS activated data across all results?
5. Could authors please label the graphs within figures with A,B ,C or similar (eg Figure 2)

Experimental design

Could the authors please comment on the choice of rats for tolerant vs susceptible groups. Why in particular 4min and 80 sec cut-offs were chosen?

Validity of the findings

Could authors please discuss the possible causes why: “Nos2 mRNA levels increased under LPS stimulation in both groups, albeit no dynamics in the iNOS protein content were revealed by Western blot analysis (Table 1, Figure 3).”

·

Basic reporting

The manuscript of Dzhalilova et al. titled "Molecular and phenotypic distinctions of macrophages in tolerant and susceptible to hypoxia rats" provides a comparative study of the differences in the inflammatory response and immunity reaction observed in bone marrow-derived macrophages collected from rats with different tolerance to hypoxia and analyzed both in non-activated and LPS-activated conditions. This study deeply investigates the molecular and phenotypic characteristics by observing the levels of several genes and some of the respective proteins and I recommend it for publication after having addressed some points:

Although Table 4 summarizes the data obtained, it is necessary to reorganize the figures and tables. It would be better to put together Figures 2 and 6, moving iNOS relative expression from Figure 2 and Figure 5 into Figure 3 and creating a panel for Figure 7 and Table 3.

Line 381: please delete "23".

Experimental design

In the classification of hypoxia-tolerant and hypoxia-susceptible rats, there is no reference for the cut-offs of 4 min and 80 s. Could the authors explain their choice?

Validity of the findings

From line 362: Citing literature, authors stated that HIF-2 induces directly VEGF and VEGF receptors expression under hypoxic conditions with a stronger transactivation capacity toward VEGF promoter compared to HIF-1 but in non-activated hypoxia-susceptible macrophages there is no correspondence between the higher level of hif2 mRNA and those of vegf and VEGF which, on the contrary, are significantly lower compared to non-activated hypoxia-tolerant macrophages. Could the authors explain this?

From line 386: From the literature that the authors cited it was demonstrated that HGF-Met binding triggers a cascade involving PI3K that activates HIF-1 and this is consistent with the results obtained from the authors since LPS-activated hypoxia-susceptible macrophages have higher HGF, hif1a, and HIF-1a levels. In addition from previous studies, it was also demonstrated that HIF-1a promotes an M1 polarization which is in contrast with the plausible role of HGF as M2 polarization agonist. Please, can the author better discuss this?

---

## Round 0.2 · accepted · Accept

I really appreciated how the authors have addressed properly all the reviewers' comments.

The manuscript is ready for publication.

·

Basic reporting

no comments

Experimental design

no comments

Validity of the findings

no comments

Additional comments

The authors have addressed all my comments. Namely they have updated and reordered the figures, amended the methods section as well as added certain explanations to the discussion section.